# Evaluation of a Robust Control System in Real-World Cable-Driven Parallel Robots

**Damir Nurtdinov & Alexei Kornaev**
Research Center of the Artificial Intelligence Institute
Innopolis University
Innopolis, Russia
`d.nurtdinov@innopolis.university` & `a.kornaev@innopolis.ru`

**Aliaksei Korshuk**
Innopolis University & Coframe
Innopolis, Russia & California, USA
`a.korshuk@innopolis.university`

**Alexander Maloletov**
Innopolis University & Volgograd State Technical University
Innopolis & Volgograd, Russia
`a.maloletov@innopolis.ru`

## Abstract

This study evaluates the performance of classical and modern control methods for real-world Cable-Driven Parallel Robots (CDPRs), focusing on underconstrained systems with limited time discretization. A comparative analysis is conducted between classical PID controllers and modern reinforcement learning algorithms, including Deep Deterministic Policy Gradient (DDPG), Proximal Policy Optimization (PPO), and Trust Region Policy Optimization (TRPO). The results demonstrate that TRPO outperforms other methods, achieving the lowest root mean square (RMS) errors across various trajectories and exhibiting robustness to larger time intervals between control updates. TRPO's ability to balance exploration and exploitation enables stable control in noisy, real-world environments, reducing reliance on high-frequency sensor feedback and computational demands. These findings highlight TRPO's potential as a robust solution for complex robotic control tasks, with implications for dynamic environments and future applications in sensor fusion or hybrid control strategies.

## 1 Introduction

Cable-Driven Parallel Robots (CDPR) have unique parameters, which means they can move heavy loads within a fairly large space. Cable suspended robots can be divided into two types: fully constrained configuration, which does not allow a free position and orientation movements of the end-effector; underconstrained configuration, which does not fully restrict the robot.

This study focuses on a real-world Cable-Driven Parallel Robot (CDPR), an under-constrained physical system, operating under a control system with limited time discretization. Our goal is to conduct a comparative analysis of classical and modern control methods and to optimize their time discretization.

## 2 Related Work

### 2.1 Classical Approach

Proportional-Integral-Derivative (PID) controllers are widely used in robotic systems, including CD-PRs, due to their simplicity and effectiveness. PID controllers are straightforward to implement and tune, making them accessible for various CDPR applications. In addition, PID controllers do not require a precise mathematical model of the system, which makes them suitable for complex systems such as CDPRs.

### 2.2 Reinforcement Learning control approach

The Actor-Critic Reinforcement Learning (RL) algorithm is a powerful approach to control CDPR. This method combines the strengths of both value-based and policy-based RL techniques.

Deep Deterministic Policy Gradient (DDPG) is an off-policy reinforcement learning algorithm designed for continuous action spaces, making it particularly suitable to control CDPR. DDPG combines the strengths of the Deep Q-Networks (DQN) and deterministic policy gradients to handle the complexities of continuous control tasks (Nomanfar & Notash, 2024).

Trust Region Policy Optimization (TRPO) is an advanced reinforcement learning algorithm designed to address the challenge of stable and efficient policy updates in complex control tasks. The core idea behind TRPO is to maximize the expected return of the policy while constraining the change in the policy at each iteration (Schulman et al., 2015).

Proximal Policy Optimization (PPO) is a popular reinforcement learning algorithm developed by OpenAI that has shown impressive performance in various control tasks, including those applicable to CDPR. PPO is designed to be simple to implement, sample efficient, and capable of solving a wide range of continuous control problems Schulman et al. (2017).

## 3 Methodology

### 3.1 Control Strategies

Various control strategies have been explored for CDPRs, including classic PID controllers which tuned with intelligent algorithms (Kel, 2005) and Reinforcement Learning (RL) that has emerged as a promising approach for CDPR control and trajectory planning (Bouaouda et al., 2024).

One of the most traditional approaches is the use of classic PID controllers, which have been tuned with intelligent algorithms to enhance performance in dynamic environments. PID controllers are favored for their simplicity and effectiveness in achieving desired positions by continuously adjusting the control inputs based on the error between the desired and actual states. Recent studies have demonstrated the effectiveness of PID controllers in managing cable tensions and ensuring precise movement of the end effector in CDPR applications (Bayani et al., 2015).

In addition to traditional control methods, RL has emerged as a promising approach to CDPR control and trajectory planning. RL algorithms, such as the Deep Deterministic Policy Gradient (DDPG) and Proximal Policy Optimization (PPO), leverage the principles of trial-and-error learning to optimize control policies based on feedback from the environment. These algorithms enable the robot to learn complex behaviors through interactions, making them particularly suitable for dynamic tasks where traditional control methods may struggle (Nomanfar & Notash, 2023).

### 3.2 Environment

The environment for the CDPR is implemented as a custom OpenAI Gym environment, providing a standardized interface for reinforcement learning algorithms. This environment encapsulates the dynamics and control of a 4-cable CDPR system. [1]. The observation space is defined as a 12-dimensional continuous space when using target velocity, or a 9-dimensional space without it. It

---

[1] https://github.com/damurka5/RL_CDPR

includes the current position, velocity, target position, and optionally, the desired velocity of the end-effector.

The action space defines control inputs for the four cables, either continuous (normalized forces between -1 and 1) or discrete (specified levels). The step function applies actions, updates the system state using state-space dynamics, and computes rewards based on target distance, proximity, and optionally, velocity. The reset function initializes each episode by randomizing start and target positions, setting velocity to zero, and resetting internal counters.

## 4 Kinematics and Dynamics

Studies have shown that for a CDPR with four cables and non-elastic sagging cables, if the ideal cable model has a single forward kinematic solution, the sagging cable model will also have a single solution (Merlet, 2021). The kinematics equation for the vector that describes $i^{th}$ cable:

$$l_i = c - a_i + R \times b_i \qquad (1)$$

where $c$ is a coordinate of end effector in world coordinate frame, $a_i$ is a cable vanishing point from the $i^{th}$ guide roller, $R$ is a rotation matrix which represents an orientation of the end effector, $b_i$ is a cable connection point. For the first approximation $b_i = \vec{0}$. Jacobian can be calculated as a unit vector $\vec{S_i}$ along $i^{th}$ cable:

$$J = \begin{bmatrix} \vec{S_1} & \vec{S_2} & \vec{S_3} & \vec{S_4} \end{bmatrix}^T \qquad (2)$$

## 5 Results and Discussion

The following section presents a comparative analysis of the performance of reinforcement learning algorithms DDPG, PPO, and TRPO implemented using the Stable Baselines3 library [2] along with a traditional PD controller, highlighting their effectiveness in controlling cable-driven parallel robots under varying conditions.

We worked with an underconstrained robot configuration that has four cables attached to the servo-motors on a fixed $2.31m \times 2.81m$ frame, the anchor points are $3.22m$ high. Each cable is connected to the box-shaped end effector and a servomotor drum through the pulley.

We created a force control mechanism to run CDPR on different trajectories. We considered the end effector as a point mass of 1 kg, drums and pulleys with zero inertia. After that we have made an environment based on Gymnasium Python classes [3], the reward is calculated from two components: the distance improvement term multiplied by 50 (empirical investigation) and the normalized proximity term multiplied by 5. The training process for all RL algorithms was implemented on a model with $\Delta t = 0.1$ sec.

### 5.1 DDPG

The DDPG agent is initialized with a multilayer perceptron (MLP) policy and configured with customizable hyperparameters such as learning rate, buffer size, batch size, and discount factor. A cosine learning rate schedule with warmup is employed to adaptively adjust the learning rate during training.

The training process for the DDPG algorithm in the CDPR environment demonstrated significant improvements in agent performance over 1.8 million episodes. Initially, the average episode length increased to approximately 28 steps per episode. This growth indicates that the agent was learning to sustain its control actions effectively to achieve better outcomes. However, as training progressed, the episode length converged to an average of 22 steps per episode.

In terms of rewards, the agent's performance improved steadily throughout the training process. By the end of training, the average episode reward reached up to 2000. The increasing reward trend indicates that the agent successfully learned to navigate the complex dynamics of the environment and optimize its control strategy.

---

[2] https://stable-baselines3.readthedocs.io/en/master/
[3] https://gymnasium.farama.org/

## 5.2 PPO

The PPO agent, as it was in DDPG, is initialized with a multilayer perceptron (MLP) policy and configured with customizable hyperparameters. A cosine learning rate schedule with warm-up was also employed.

The training process for the Proximal Policy Optimization (PPO) algorithm demonstrated significant differences between continuous and discrete action spaces over 7000 episodes. For continuous PPO, the average episode length converged to approximately 22 steps per episode. In contrast, the discrete PPO initially increased to about 21 steps per episode before converging to a lower average of 12 steps. The reward outcomes further underscore the advantage of the discrete action space. While the continuous PPO achieved an impressive episode reward of up to 3000 by the end of training, the discrete PPO significantly outperformed it with rewards reaching approximately 7500. This substantial difference in reward accumulation highlights the discrete PPO's superior ability to optimize the control policy for the CDPR system, resulting in more precise and efficient movements that better satisfy the task objectives.

## 5.3 TRPO

We used the same training process technique for the Trust Region Policy Optimization (TRPO) algorithm, as well as we used the cosine learning rate schedule with warmup.

The training process for the TRPO algorithm demonstrated impressive performance over 6000 episodes. The average episode length converged to approximately 10 steps per episode. This low number of steps indicates that TRPO quickly learned an efficient control strategy for the CDPR system. The episode reward reached up to 10000 by the end of the training process, which is a significant improvement compared to the results observed for other algorithms like PPO and DDPG. This high reward value indicates that TRPO not only learned to complete the task quickly but also with high precision and efficiency. The combination of low average episode length and high reward suggests that TRPO developed a policy that could accurately control CDPR while minimizing unnecessary movements and optimizing the path to the target position.

After the main training process, we implemented two key changes on pre-trained model for TRPO algorithm: the addition of initial velocity to the state representation and prioritizing initial points closer to the target position. As a result of these training adjustments, the TRPO model was able to overcome the initial learning hurdles and develop a more robust control policy for the CDPR system.

The results of the best-performed reinforcement learning algorithms are shown in Fig. 1. This figure illustrates performance metrics, such as the mean episode length (a) and reward accumulation (b). Evaluation of the reinforcement learning algorithms and PID controller on three different trajectories

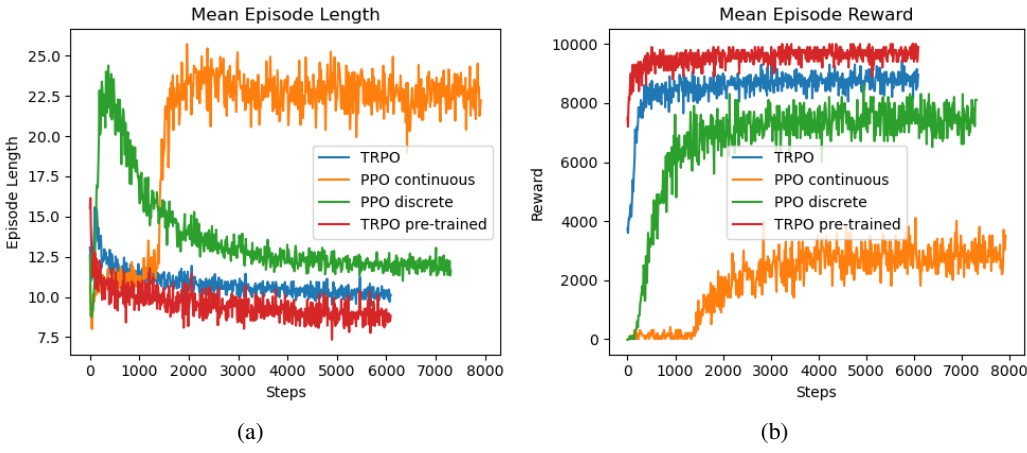

(a)                                    (b)

Figure 1: training metrics

Table 1: RL models and PID controller RMS Errors [m]

| Trajectory | DDPG | PPO | TRPO | PID |
|---|---|---|---|---|
| Circle | 0.0098 | 0.0113 | 0.0075 | 0.0489 |
| Spiral 1 | 0.0235 | 0.0125 | 0.0079 | 0.0149 |
| Spiral 2 | 0.0114 | 0.0133 | 0.0084 | 0.0167 |

is shown in Table 1. TRPO consistently achieves the lowest errors, outperforming the other models and the PID controller on all trajectories.

## 5.4 MODEL SIMULATIONS ON DIFFERENT $\Delta t$

We have conducted several experiments to check the sustainability and robustness of control strategies on different simulation time intervals, and used optimal gains for PID controller for each experiment. The results shown in Fig. 2 show that RL algorithm has learned the robot's behavior and can

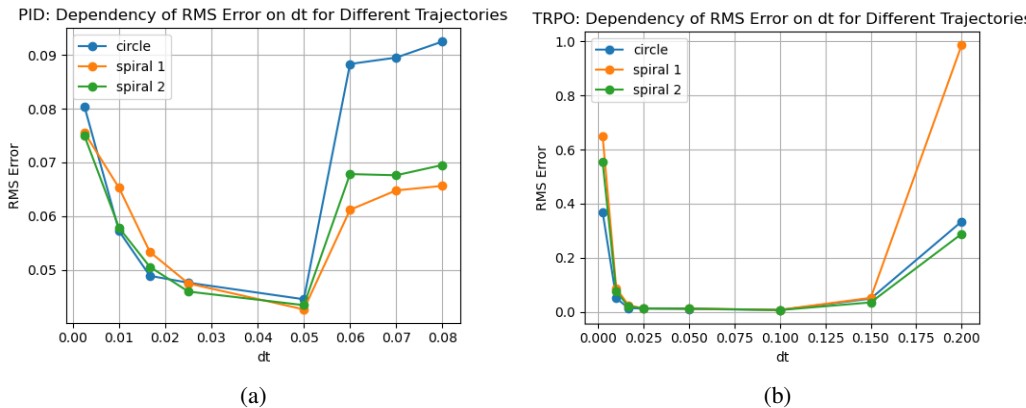

(a)                                    (b)

Figure 2: different $\Delta t$ simulations

operate at larger $\Delta t$ values than a usual PID controller.

## 6 CONCLUSION

The TRPO algorithm demonstrated superior performance compared to DDPG and PPO in both continuous and discrete settings, showcasing its potential for complex robotic control tasks. Its stability with larger time intervals ($\Delta t$) makes it particularly suitable for real-world cable-driven parallel robots, where sensor noise and latency limit high-frequency precision. Unlike PID controllers, TRPO reduces computational demands and adapts to real-world imperfections, making it a strong candidate for dynamic environments. Future work will include physical experiments on real robots to further validate these findings and enhance their practical applicability.

## 7 ACKNOWLEDGEMENT

All authors were supported by the Research Center of the Artificial Intelligence Institute of Innopolis University.

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
