# OpenReview forum: "Evaluation of a Robust Control System in Real-World Cable-Driven Parallel Robots"
_ICLR.cc/2025/Workshop/AgenticAI — ICLR 2025 Workshop AgenticAI Poster_

### Official Review · Reviewer_4PmK · 2025-03-01
**Strong Analysis and Reproducibility, but Lacks Real-World Validation and Efficiency Insights**

**Rating:** 6
**Confidence:** 4

**Review:**

Summary:
This paper evaluates the performance of classical PID controllers and modern reinforcement learning algorithms for controlling Cable-Driven Parallel Robots (CDPR) in real-world conditions. The study highlights TRPO's stability, efficiency, and adaptability in dynamic environments, positioning it as a promising approach for advanced robotic control applications.

Strengths:
1. The study systematically evaluates classical PID controllers alongside state-of-the-art reinforcement learning techniques. This comparative analysis offers valuable insights into the strengths and weaknesses of both approaches in CDPR control.
2. Unlike previous works that focus on ideal simulation conditions, this study examines the impact of time discretization on control performance. The results indicate that TRPO can maintain stability even with lower sensor update rates, making it more applicable to real-world deployment.
3. The authors have released their code for reproducibility.

Weaknesses:
1. While the study is based on real-world CDPRs, the evaluation is entirely simulation-based. Adding physical experiments would significantly enhance credibility.
2. TRPO requires significantly higher computational resources than PID controllers or even DDPG/PPO. However, the paper does not discuss the training time, hardware requirements, or inference speed. This information is crucial for assessing the practical feasibility of deploying TRPO in real-time control systems.
3. The paper provides limited insight into how hyperparameter tuning affects RL performance. An ablation study on learning rates, exploration strategies, and reward function variations would be valuable.

---

### Official Review · Reviewer_ajxj · 2025-03-02
**Review of Robust Control System**

**Rating:** 6
**Confidence:** 4

**Review:**

# Summary:
This work compares classical PID controllers with modern RL algorithms for CDPRs. The results show that TRPO outperforms the other methods, achieving the lowest RMS and demonstrating robustness to larger control update intervals. These findings suggest TPRO as a promising solution for complex robotic control tasks, with potential applications in various real-world environments.

# Strengths:

1. The work is well-organized and easy to follow.
2. The study highlights TRPO's various advantages, providing valuable insights into its effectiveness in robust control system.
3. This work conducts extensive experiments to thoroughly evaluate the performance of different control methods, providing a comprehensive analysis of their effectiveness in real-world scenarios.

# Weaknesses:

1. The paper has a relatively limited coverage of related work.

---

### Decision · Program_Chairs · 2025-03-05

Accept (Poster)